# The Cost-Effectiveness of Sentinel Lymph Node Mapping in High-Risk Endometrial Cancer

**DOI:** 10.3390/cancers16244240

**Published:** 2024-12-19

**Authors:** Lara C. Burg, Robin J. Vermeulen, Ruud L. M. Bekkers, Roy F. P. M. Kruitwagen, Petra L. M. Zusterzeel

**Affiliations:** 1Department of Gynaecological Oncology, Radboud University Medical Center, 6525 GA Nijmegen, The Netherlands; ruud.bekkers@catharinaziekenhuis.nl (R.L.M.B.); petra.zusterzeel@radboudumc.nl (P.L.M.Z.); 2Department of Medical Imaging, Radboud University Medical Center, 6525 GA Nijmegen, The Netherlands; robin.vermeulen@radboudumc.nl; 3Department of Obstetrics and Gynaecology, Catharina Hospital, 5623 EJ Eindhoven, The Netherlands; 4Department of Obstetrics and Gynaecology, Maastricht University Medical Center, 6229 HX Maastricht, The Netherlands; r.kruitwagen@ziggo.nl; 5GROW—School for Oncology and Reproduction, Maastricht University, 6229 ER Maastricht, The Netherlands

**Keywords:** endometrial cancer, high risk, sentinel lymph node mapping, lymphadenectomy, indocyanine green, cost-effectiveness analysis

## Abstract

This study examined the cost-effectiveness of sentinel lymph node mapping versus routine pelvic lymphadenectomy for assessing lymph nodes in patients with high-risk endometrial cancer. Using a decision analytic model, we compared the two methods based on costs and health outcomes. We found that sentinel lymph node mapping was more effective and less expensive than lymphadenectomy. The cost savings and improved outcomes primarily stemmed from the fewer side effects associated with sentinel lymph node mapping. The analysis confirmed that sentinel lymph node mapping is the best option for determining the need for additional treatment in these patients. Overall, this study suggests that sentinel lymph node mapping should be the preferred method for lymph node assessment in people with high-risk endometrial cancer.

## 1. Introduction

With health care costs rising drastically over time, a critical assessment of both the costs and effects of new medical techniques is important before implementation [1]. Endometrial cancer (EC) is the most common gynecological cancer in high-income countries and is associated with high healthcare costs. The cost-effectiveness of the treatment for EC patients is therefore important.

EC is differentiated into low-grade endometrioid, high-grade endometrioid, and non-endometrioid tumors. A further risk stratification is based on clinical and pathological factors such as the patients’ age, stage, myometrial invasion, histology, and molecular profile. Grade 3 endometrioid EC and non-endometrioid EC are usually classified as high-risk EC and account for approximately 20% of all people with EC [2]. Patients with high-risk EC have an increased risk of metastases, especially lymph node metastases [3]. Lymph node metastasis is an important prognostic factor and determines the needed type of adjuvant therapy. Assessing the lymph node status remains a subject of debate, with high variance in clinical practice among gynecological oncologists worldwide. Since adjuvant treatment is based on the surgical stage and thus the lymph node status, different lymph node assessment strategies are related to different rates of use in adjuvant therapy, having subsequent impacts on prognosis, resulting in different rates of treatment-specific side effects [4,5]. Historically, routine pelvic and para-aortic lymphadenectomy comprised the standard approach to assessing lymph node status. Sentinel lymph node (SLN) mapping is a minimally invasive alternative and is nowadays supported by multiple international organizations (ESGO/ESTRO/ESP) [6,7]. Multiple prospective studies have shown the feasibility of SLN mapping [8]. Additionally, SLN mapping leads to less morbidity such as lymphedema compared to lymphadenectomy [9]. However, in multiple guidelines including the Dutch EC guideline, SLN mapping is not mentioned yet as an alternative to lymphadenectomy [2]. To ascertain the optimal strategy concerning health outcomes, it is essential to evaluate the consequences of various strategies alongside their associated costs. Consequently, the objective of this decision–analytic modeling study was to compare the cost-effectiveness of sentinel lymph node (SLN) mapping with that of routine lymphadenectomy in patients diagnosed with high-risk endometrial cancer (EC).

## 2. Methods

### 2.1. Patient Population

The target population comprised patients who were preoperatively diagnosed with presumed early-stage EC, with grade 3 endometrioid, serous, or clear cell histology (high–intermediate- or high-risk EC).

### 2.2. Decision Model

A decision–analytic model was developed to assess the cost-effectiveness of sentinel lymph node (SLN) mapping in comparison to that of routine lymphadenectomy for guiding the selection of adjuvant therapy. A decision tree represented the clinical pathway associated with each diagnostic strategy, and a Markov model simulated long-term follow-up in terms of survival, health-related quality of life (HRQOL), and healthcare costs. 

Figure 1 illustrates the decision tree for both strategies. All patients underwent a minimally invasive hysterectomy with bilateral saplingo-oophorectomy. In the first strategy, pelvic lymphadenectomy was incorporated as part of the standard surgical procedure (including lymphatic tissue on the external iliac vessels anteriorly and medially, over the internal iliac vessels, at the interiliac junction, and over the obturator nerve). It was assumed that lymphadenectomy served as the gold standard for detecting lymph node metastases; therefore, the model did not account for false negative or false positive results. The second strategy involved sentinel lymph node (SLN) mapping using a cervical injection of indocyanine green (ICG) to evaluate lymph node metastases. SLN mapping was performed in conformance with the standard guidelines. No extra costs were incorporated for the use of a near-infrared (NIR) camera, as the commonly used laparoscopy devices are equipped with NIR and therefore are available without extra costs (for example, the Firefly ™ Fluorescence Imaging for da Vinci^®^ Xi ™, D-Light P system for Storz, near-infrared system for Olympus). Other SLN mapping techniques, such as the use of patent blue dye with scintigraphy, were not included in this analysis, as ICG is the preferred technique. ICG shows a comparable or even better detection rate, is easier in use, and is less expensive [6]. In the model, in cases of unilateral or bilateral failure of SLN mapping, a side-specific lymphadenectomy was performed. No differences in costs for hospital stay were calculated between the lymphadenectomy group and the SLN mapping group, as the common postoperative stay in both groups was one day.

In case of detected lymph node metastases in both strategies, patients received adjuvant therapy. Multiple adjuvant strategies are performed nationally and internationally, varying from external beam radiation therapy (EBRT) or chemotherapy alone to a combination of EBRT and chemotherapy. We therefore decided to run the model three times: (1) patients with lymph node metastases were treated with EBRT only; (2) patients with lymph node metastases were treated with chemotherapy only; (3) patients with lymph node metastases were treated with a combination of EBRT and chemotherapy. If neither lymphadenectomy nor SLN mapping showed lymph node metastases, patients were adjuvant-treated with vaginal brachytherapy (VBT).

The Markov model included several potential health states: post-treatment (the status following primary surgery and adjuvant therapy), recurrence, and death (Figure 2). The post-treatment health state was subdivided into three categories based on the treatment received and the presence or absence of lymph node metastases, accounting for the small risk of undetected positive lymph nodes in the sentinel lymph node (SLN) strategy. The recurrence health state encompassed both pelvic and distant recurrences. A minor proportion of patients could experience vaginal recurrence, of which the occurrence and consequences were considered in the post-treatment health state. In instances of pelvic or distant recurrence, patients received chemotherapy as salvage treatment. Death could occur due to cancer-related causes or other factors (background mortality). The cycle duration in the Markov model was one year, since this aligned well with the natural disease progression. After each annual cycle, patients had a probability of transitioning between health states. For each year spent in a specific health state, patients were assigned outcomes related to health-related quality of life (HRQOL) and the costs associated with that state. We assumed that each year, patients would have two follow-up visits, reflecting the mean number of visits based on the follow-up schedule recommended in clinical guidelines [2]. Patients in the recurrence state were assigned a health state utility value corresponding to regional failure. The model was simulated over a total duration of twenty years, equating twenty cycles [10,11]. This duration was selected as women are typically diagnosed with endometrial cancer (EC) at a mean age of 65 years, and the average life expectancy for women at this age is approximately 20 years [10]. 

### 2.3. Probability Decision Tree

In the target population of people with preoperative, early-stage, grade 3 endometrioid, serous or clear cell EC, the probability of 21% of having lymph node metastases was based on pooled data [11,12,13]. No correction for isolated para-aortic lymph node metastases was applied, as they occur in only 1% of patients [14]. The probability of bilateral SLN mapping was 79% [6]. The sensitivity of 97% for SLN mapping was based on a study by Rossi et al. [15]. The specificity was assumed to be 100%, since no false positive outcomes are possible with SLN mapping. Lymph nodes with isolated tumor cells were excluded. The probability of short-term surgical and treatment-related side effects were derived from the literature. Both lymphadenectomy and SLN mapping are supposed to be diagnostic and therefore have no direct impact on the probability of developing recurrent disease. An overview of the input values is shown in Table 1.

### 2.4. Probability Markov Model

Patients moved between different health states in the Markov part of the model according to a set of transition probabilities (Table 1). Transition probabilities and probabilities of developing long-term surgical- and adjuvant-treatment-related side effects were derived from the literature, as depicted in Table 1. We assumed that all people who died of cancer-related causes first experienced regional and/or distant recurrence. The overall failure rate, which includes regional and distant recurrences, was based on clinical trials [4,5,16,17,18]. Mortality rates included all-cause mortality, both cancer- and non-cancer-related (background mortality). The proportion of people that died from causes other than cancer-related causes was subtracted from the overall mortality rate and was not assigned the costs or consequences of salvage treatment for recurrent disease in the model. Background mortality was based on Dutch statistics on age-related death among women in the general population [10].

### 2.5. Costs

The cost analysis was performed from a Dutch healthcare perspective and costs included in the model were in Euros (EUR). Costs per minute of surgery, including personnel, were based on a bottom-up cost analysis for operating rooms, while surgery time was based on the literature and expert opinion (three gynecological oncologists affiliated with three different tertiary centers; Table 1) [19,20,21]. The costs for VBT and EBRT were based on diagnosis treatment combination prices from the Dutch healthcare authority, and costs for chemotherapy were based on reference prices from the Dutch pharmacotherapeutic compass [22]. The costs for the treatment of initial and yearly lymphedema and lymphocele treatment were based on the costs specified by the National Lymphedema Network [23]. No differences in follow-up duration or the number of hospital days following initial treatment were anticipated between the strategies. In the event of disease recurrence, the costs associated with follow-up care were incorporated into the analysis [24]. Costs were indexed to 2023 prices. The annual discount rate for costs was set at 4% according to the Dutch guideline [25]. An overview of the costs is shown in Table 1.

### 2.6. Effectiveness Measures

The effectiveness of the strategies is expressed as quality-adjusted life years (QALY), a measure combining survival and quality of life. Quality of life is expressed on a health state utility scale ranging from 0 to 1, representing death and perfect health, respectively [26]. Utility data were derived from the studies of Einstein et al., Chen et al., and Havrilesky et al., and the applicability of these data for our model was discussed with experts in the field (three gynecological oncologists affiliated with three different tertiary centers) [27,28,29]. Patients without any side effects or recurrent disease were assumed to be in perfect health and were assigned a utility value of 1. In the case of side effects, a disutility value was subtracted in each health state for the proportion of patients suffering from this side effect. An overview of utilities for each health state is provided in Table 1. The annual discount rate for effects was 1.5% [24].

### 2.7. Validity Testing

The model was validated according to the AdViSHe checklist, by means of consulting clinical experts, extreme value testing, subunit testing, and cross-validation with the relevant literature [30].

### 2.8. Data Analysis

The costs and effects associated with both strategies were compared, and an incremental cost-effectiveness ratio (ICER) was calculated, where applicable. The ICER reflects the additional costs required to achieve one extra quality-adjusted life year (QALY) when comparing one strategy to another. A strategy is considered cost-effective if the ICER is below the established willingness-to-pay threshold for a QALY. In this study, a willingness-to-pay threshold of EUR 20,000 per QALY was assumed, representing the lowest reference value for such thresholds in the Netherlands. The model was run three times, with each run applying a different adjuvant treatment strategy for patients with lymph node metastases: either external beam radiotherapy (EBRT) alone, chemotherapy alone, or a combination of EBRT and chemotherapy. A one-way deterministic sensitivity analysis was conducted to assess the sensitivity of the model to variations in individual parameter values. Input values, including surgical time, probability of bilateral mapping, sensitivity of SLN mapping, transition probabilities, probabilities of side effects, prevalence of lymph node metastasis, utility values, and costs, were manually varied over a range of 20%. To evaluate the impact of uncertainty over all input values on the outcomes, a probabilistic sensitivity analysis was performed with 1000 iterations for each parameter (Table 1). Model development and analysis were executed in R (version 4.2.3, the R Foundation for Statistical Computing, Vienna, Austria), utilizing code examples provided by the Decision Analysis in R for Technologies in Health workgroup [31].

**Table 1 cancers-16-04240-t001:** Input parameters: clinical parameters, cost estimates, and utility values.

Parameter	% (n)	95% CI	Source
**Probabilities**			
Prevalence metastases	21%	20–23%	[11,12,13]
*SLN detection*			
Bilateral	79% (179/227)		
Unilateral	16% (37/227)		[6]
Failed	5% (11/227)		
Sensitivity SLN	97% (35/36)		[15]
*Operation time*			
Unilateral SLN	10 min	8–12 min	[20,32]
Unilateral lymphadenectomy	30 min	24–46 min	[20]
*Short-term side effects*			
EBRT	44% (145/326)		[33]
Chemotherapy	90% (325/361)		[5]
EBRT + chemotherapy	96% (332/346)		[5]
VBT	21%	12–34%	[34]
Lymphadenectomy			
Lymphedema	19%	13–18%	[31,32,35,36,37,38,39,40,41,42,43,44,45,46]
Symptomatic lymphocele	6%	4–10%	[35,41,45,46,47,48]
SLN mapping			
Lymphedema	5%	0–9%	[20]
Symptomatic lymphocele	2%	1–3%	Expert opinion
*Long-term side effects*			
EBRT	23% (43/187)		[4]
Chemotherapy	35% (125/361)		[5]
EBRT + chemotherapy	38% (76/201)		[4]
VBT	26%	19–34%	[34]
Lymphadenectomy			
Lymphedema	19%	13–18%	[31,32,35,36,37,38,39,40,41,42,43,44,45,46]
SLN mapping			
Lymphedema	5%	0–9%	[20]
*5 year failure rate*			
Vaginal recurrence			
LNM+ and EBRT	2.10%	1.0–4.4%	[4]
LNM+ and chemotherapy	4.9% (18/348)		[5]
LNM+ and EBRT + chemotherapy	1.9% (7/370)		[5]
LNM+ and VBT	67% (10/15)		[16]
LNM− and VBT	6.1% (22/363)		[17]
Regional (pelvic) and/or distant			
recurrence			
LNM+ and EBRT	41.60%		[4]
LNM+ and chemotherapy	42%	34–50.2%	[5]
LNM+ and EBRT + chemotherapy	29.10%	36–47%	[4]
LNM+ and VBT	67% (10/15)	22.6–37.1%	[16]
LNM− and VBT	25%		[18]
*5 year mortality rate*			
LNM+ and EBRT	31.50%	23.3–38.8%	[4]
LNM+ and chemotherapy	23.5% (86/366)		[5]
LNM+ and EBRT + chemotherapy	21.50%	14.6–27.8%	[4]
LNM+ and VBT	70%	60–80%	Expert opinion
LNM− and VBT	6.10%		[17]
**Costs**			
Histopathology lymphadenectomy	823.71	658.97–988.45	Radboudumc cost data
Histopathology SLN	1189.02	951.22–1426.82	Radboudumc cost data
*Unilateral SLN mapping*			
OR time + medical specialists	11.08/min	7.1–11.8	[21]
Indocyanine green	35.18	24–36	Expert opinion
*Unilateral lymphadenectomy*			
OR time + medical specialists	11.08/min	7.1–11.8	[21]
*Complications*			
Lymphedema treatment			
Initial	1114	760–1140	
Yearly	2579.78	1760–2640	Lymphedema network
Lymphocyst drainage	934.59	637.6–1147.68	
EBRT	8237.72	6092–9138	Dutch Healthcare authority
Chemotherapy	8682.17	5923.2–8884.8	Pharmacotherapeutic compass
VBT	9269.63	6.572–9858	Dutch Healthcare authority
Standard follow-up	138	88.0–133.2	Dutch Healthcare authority
**Utilities**			
No recurrence, no toxicity	1	Fixed	[28] and expert opinion
Radiotherapy induced toxicity	0.866	0.758–0.946	[28] and expert opinion
Chemotherapy induced toxicity	0.50 (SD 0.21)		[29]
Combined chemo and radiotherapy induced toxicity	0.50 (SD 0.21)		[29]
Lymphedema	0.925	0.851–0.998	[27] and expert opinion
Vaginal recurrence	0.69	0.58–0.79	[28] and expert opinion
Regional/distant recurrence	0.38	0.30–0.45	[28] and expert opinion
Death	0	Fixed	[28] and expert opinion

LNM = lymph node metastasis.

## 3. Results

### 3.1. Base Case Analysis

SLN mapping was the most effective strategy in each run (i.e., independent of the type of adjuvant therapy; Table 2). The strategy of SLN mapping followed by concomitant EBRT and chemotherapy as the standard treatment for patients with lymph node metastases resulted in the highest effectivity, with 11.76 QALYs expected after 20 years. The least effective strategy was performing lymphadenectomy followed by chemotherapy, which resulted in 11.49 QALYs expected at the end of the total model duration.

SLN mapping was less expensive compared to lymphadenectomy, with a difference of approximately EUR 3500–3600 in each run. As SLN mapping was less costly and more effective compared to lymphadenectomy, SLN seems to be the more appropriate strategy compared to lymphadenectomy for guiding the need for adjuvant therapy independent of the type of standard adjuvant therapy.

### 3.2. Deterministic Sensitivity Analysis

SLN mapping remained the least expensive strategy over the full range of sensitivity percentages compared to lymphadenectomy for all three options regarding adjuvant therapy. It remained the most cost-effective strategy in terms of QALYs for SLN sensitivity percentages of 91%, 89%, and 91% for EBRT, chemotherapy, and combined adjuvant treatment, respectively, with a WTP threshold of 20,000 EUR/QALY.

Deterministic variations in the probability of the bilateral detection rate, SLN sensitivity, probability of lymphedema, distant recurrence rate, histopathology costs of SLN, yearly lymphedema treatment costs, costs for EBRT, and health state utility value of lymphedema showed variations in the costs and/or QALYs that were larger than 5%. Variations in the sensitivity of SLN mapping had the strongest impact on the effects. Variations in the probability of lymphedema after lymphadenectomy, SLN bilateral detection rate, and yearly costs of lymphedema had the strongest impact on variations in costs. These variations were comparable in the model runs with the different types of adjuvant therapy. Figure 3 shows the results of the deterministic sensitivity analysis with EBRT as the type of adjuvant therapy for the variables that resulted in a minimal 5% change in incremental costs and/or effects.

### 3.3. Probabilistic Sensitivity Analysis

The incremental costs and effects of SLN mapping compared to lymphadenectomy for 1000 iterations are presented in Figure 4. With a willingness-to-pay threshold of EUR 20,000 per QALY, SLN mapping emerged as the cost-effective alternative in all iterations across each treatment strategy: EBRT, chemotherapy, or a combination of EBRT and chemotherapy as adjuvant therapy.

## 4. Discussion

SLN mapping appears to be cost-effective and even dominant compared to lymphadenectomy in the assessment of the lymph node status in high-risk EC and to guide the choice for adjuvant therapy. The probabilistic sensitivity analysis indicated that SLN mapping was cost-effective in all iterations (100%) at a willingness-to-pay threshold of EUR 20,000 per QALY. The increase in QALYs and decrease in costs for SLN mapping compared to lymphadenectomy were mainly due to a reduction in lymph node assessment-related side effects.

Comparable results on the cost-effectiveness of SLN mapping were seen in a previous paper in patients with low- and intermediate-risk endometrial cancer [49]. Although that paper shows similarities to the current one, some important differences should be highlighted. First, the previous paper concerned patients with stage 1, grade 1–2 endometrioid endometrial cancer, instead of high-risk endometrioid and non-endometrioid endometrial cancer, resulting in a different a priori risk of lymph node metastases. This impacts the weight given to side effects and to the accidental missing of lymph nodes and therefore to the accidental missing of metastases and adjuvant treatment. This is, for example, depicted by the differences in the failure rates. Furthermore, the previous article compared SLN mapping to two different strategies, namely, adjuvant therapy based on (a) postoperative risk factor assessment and (b) lymphadenectomy. It therefore evaluated both expending the surgical intervention (SLN mapping instead of risk factor assessment) and limiting the surgical intervention (SLN mapping instead of lymphadenectomy). This allowed the impact of SLN mapping to be determined for countries with different guidelines on lymph node assessment. The current study compared SLN mapping solely to lymphadenectomy, as (any kind of) surgical evaluation of lymph nodes is common practice in high-risk endometrial cancer.

Whether SLN mapping affects the long-term prognosis of patients with endometrial cancer has been of great concern. Studies have been performed to compare the oncological outcome of SLN mapping versus lymphadenectomy, showing that SLN mapping is not inferior to lymphadenectomy [50,51,52]. Moreover, prospective and retrospective studies indicated that SLN mapping does not appear to negatively impact oncologic outcomes in high-risk endometrial cancer [53]. In fact, SLN mapping is nowadays even described as superior to lymphadenectomy, since approximately 5% of SLNs are located outside the routinely dissected lymphadenectomy areas, and they would be missed during routine (pelvic) lymphadenectomy [54,55,56]. SLN mapping therefore benefits accurate staging.

Contradictive data are also available. The SEPAL study showed that para-aortic lymphadenectomy was associated with favorable survival outcomes in high-risk endometrial cancer [57,58]. These studies were, however, retrospective in design. No survival differences were noted between patients who underwent SLN mapping and those who underwent lymphadenectomy in the eight studies that compared recurrence-free survival in the systematic review by Bodurtha Smith et al. [59]. When interpreting studies on the survival outcomes of SLN mapping, other factors need to be considered as well. The recent shift in the focus of adjuvant therapy from radiotherapy to chemotherapy, due to the PORTEC-3 trial, might influence the results regarding the survival benefit [4]. If the survival outcome of SLN mapping is superior to that of lymphadenectomy, this might underscore the survival benefit of detection alone rather than resection. To ultimately assess the oncological outcomes of SLN mapping without lymphadenectomy, the ALICE trial is currently ongoing [60]. This is an open-label, noninferiority, randomized trial hypothesizing that pelvic and para-aortic lymphadenectomy does not add survival benefit compared to SLN mapping. The Korean Gynecologic Oncology Group is investigating this as well. They are performing a multicenter, single-blind, randomized controlled trial to determine the prognostic value of SLN mapping alone compared with conventional lymphadenectomy, with 3-year disease-free survival as the primary endpoint [61].

Despite the lack of high-quality evidence of oncological outcomes, for many gynecologists, SLN mapping has already replaced lymphadenectomy in high-risk EC [62]. The joint guideline by the European Society of Gynecological Oncology (ESGO), the European Society for Radiotherapy and Oncology (ESTRO), and the European Society of Pathology (ESP) recommends SLN mapping in high-risk EC, as it is an acceptable alternative to lymphadenectomy. With this cost-effectiveness study, we show that implementing SLN mapping as the standard of care in high-risk EC is a cost-effective alternative to lymphadenectomy. If SLN mapping is performed, indocyanine green with cervical injection is the preferred detection technique, and side-specific systematic lymphadenectomy should be performed if nothing is detected with SLN mapping on either pelvic side [7].

In the Netherlands, pelvic and para-aortic lymphadenectomy is still performed in people with high-risk EC. SLN mapping is not mentioned in our national guidelines, despite the literature showing it to be safe and feasible with less complications. We believe that our study is the first to assess the cost-effectiveness of SLN mapping in high-risk EC. With rising healthcare costs in mind, this might be an extra motivation to implement SLN mapping in high-risk EC. People with high-risk EC are nowadays already centralized in tertiary hospitals and treated by gynecological oncologists; it should be feasible to change their policy into SLN mapping. To provide extra oncological safety, it might be an option to perform subsequent lymphadenectomy during the learning phase of SLN mapping.

Our decision–analytic model study has some limitations that need consideration. Firstly, some input values were based on numerous references, leading to the increased heterogeneity of the population. This was partly corrected by pooling the data for some of the input values. For the sensitivity of SLN mapping, only one reference was used. However, other articles (including systematic reviews and meta-analyses) with large sample sizes show comparable diagnostic performance [6]. For the health state ‘no recurrence, no toxicity’, we assumed a utility value of one, whereas actually the mean utility for age matched participants in the general population is lower than one [63]. The utility value of one represents the best possible health state within this study’s context, even though it might slightly exceed the typical health values in the general population. Using a utility of one for this health state allowed the model to clearly capture the relative impact of being disease-free versus experiencing disease progression or treatment side effects. For some other input values, we had to make assumptions. We used opinions from experts in the field, i.e., three gynecological oncologists affiliated to three different tertiary centers, to discuss the applicability of the input data for our model and performed sensitivity analyses to evaluate the impact of the uncertainty in the input values. See Tale 1 for an overview of the parameters in which expert opinions were taken into account. The sensitivity analyses showed that the model outcomes are robust to variations in input values.

Secondly, in some of the patients who underwent SLN mapping, the lymph node status might have been unknown, which we did not incorporate into the model. A cause for an unknown lymph node status is an empty packet dissection, in which only fat tissue is dissected. This occurs in approximately 5% of procedures, and this rate decreases with the increase in the number of performed procedures [64]. The problem of empty packet dissection can be solved by performing a frozen section on the dissected specimen.

Thirdly, the choice of adjuvant treatment in our model was based on current practical national and international guidelines. However, adjuvant therapy is becoming more and more tailored based on molecular and genetic profiling, as stated by the TCGA and incorporated in the recent FIGO 2023 staging system on endometrial cancer [65,66,67]. We decided to not implement this in the model because the applicability of molecular profiling was not yet guaranteed worldwide. The same applied to the addition of immunotherapy such as pembrolizumab to standard chemotherapy, which results in significantly longer progression-free survival compared to chemotherapy alone [68].

Finally, we had to simplify some of the adjuvant treatment strategies for the sake of the model. For example, some clinicians might advise using more than VBT alone in high-risk EC without lymph node metastases. With the many different policies internationally, and even nationally, we had to make an assumption regarding the adjuvant treatment strategy to be included in the model. Additionally, we did not distinguish between regional or distant failure in the recurrence health state of the Markov part of the model. Patients with any recurrent disease had the same disutility, and the same costs for treatment (chemotherapy) were charged.

Our model offers clinically relevant insights regarding lymph node assessment in patients with high-risk endometrial cancer from a Dutch healthcare perspective. Regarding the applicability of our study to other countries: comparable results are expected for countries with comparable healthcare systems, i.e., most of the western countries. Currently, there is no international consensus on the optimal strategy for this assessment. Given that SLN mapping appears to be the cost-effective approach, it should be incorporated into clinical guidelines as the preferred method for determining the necessity of adjuvant treatment.

## 5. Conclusions

In conclusion, this study provides strong evidence that SLN mapping is the superior strategy for lymph node assessment in patients with high-risk endometrial cancer, both endometrioid and non-endometrioid types. By demonstrating greater effectiveness and lower costs compared to routine pelvic lymphadenectomy, SLN mapping emerges as a cost-effective approach for guiding adjuvant therapy. The model’s reliance on QALYs as a measure of effectiveness further substantiates the benefits associated with SLN mapping, particularly the significant reduction in the side effects related to lymph node assessment. Additionally, the robustness of the model’s outcomes, confirmed through sensitivity analyses, indicates that SLN mapping is a reliable option across various scenarios. Therefore, incorporating SLN mapping into clinical guidelines for people with high-risk endometrial cancer could enhance treatment decision making and optimize healthcare resources.

## Figures and Tables

**Figure 1 cancers-16-04240-f001:**
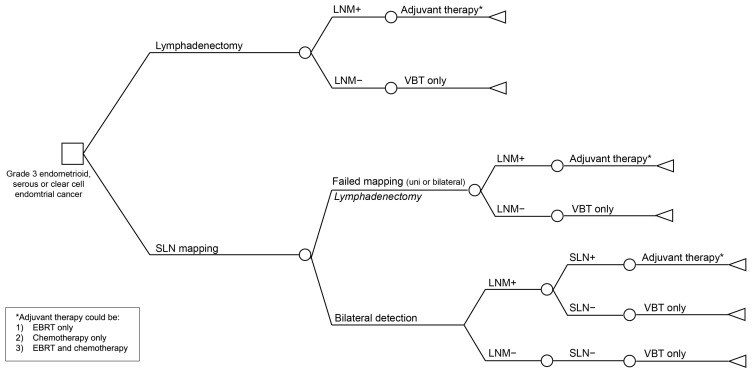
Decision tree representing the initial diagnostic and treatment pathway of both strategies. LNM = lymph node metastasis; VBT = vaginal brachytherapy; SLN = sentinel lymph node.

**Figure 2 cancers-16-04240-f002:**
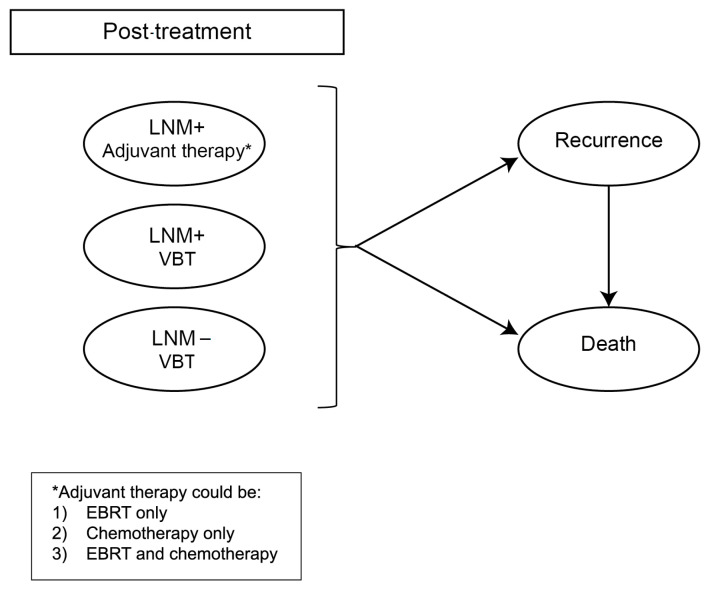
Health states of the Markov part of the model, representing the follow-up of the patients after the diagnostic pathway and the corresponding treatment.

**Figure 3 cancers-16-04240-f003:**
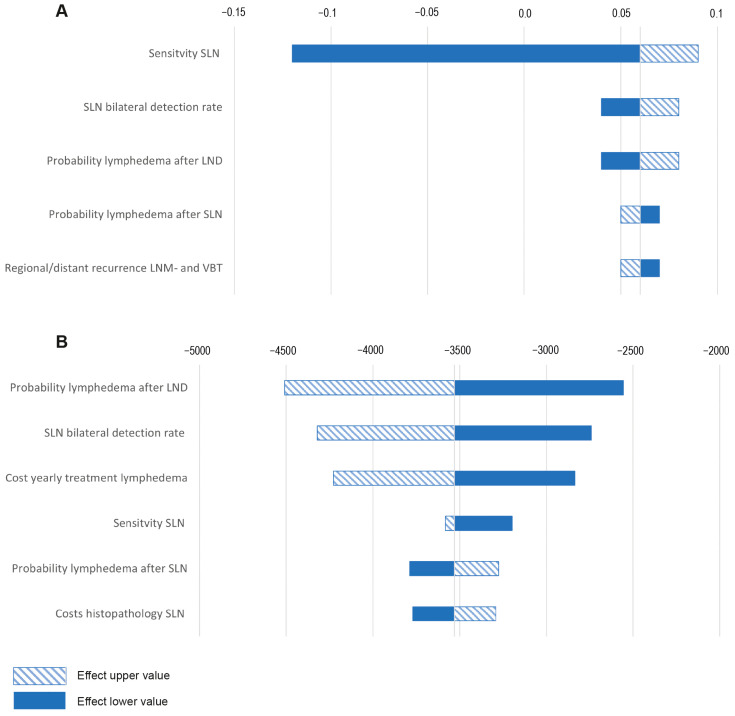
Tornado plot for deterministic sensitivity analysis. (**A**) Effects; (**B**) costs.

**Figure 4 cancers-16-04240-f004:**
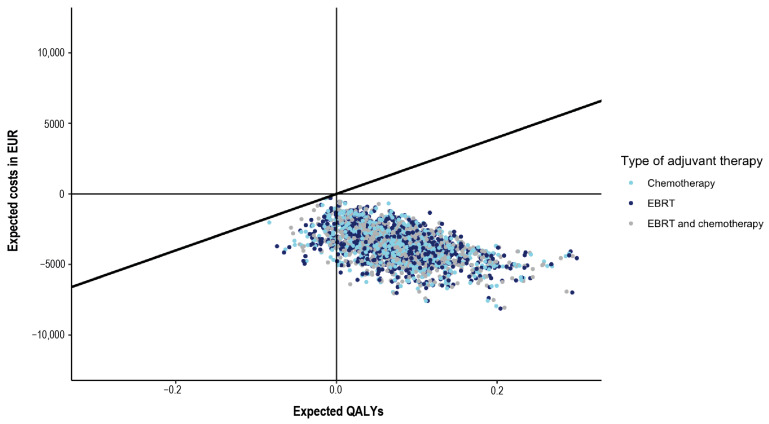
Incremental cost-effectiveness plane.

**Table 2 cancers-16-04240-t002:** Incremental analysis.

Strategy	Costs (EUR)	Effects (QALY)	Incremental Costs (EUR)	Incremental Effects (QALY)	Outcome
A: EBRT as standard treatment for patients with lymph node metastases.					
Lymphadenectomy	20,773.78	11.63	NA	NA	-
SLN mapping	17,242.10	11.69	−3531.67	0.06	Dominant
B: Chemotherapy as standard treatment for patients with lymph node metastases.					
Lymphadenectomy	20,856.08	11.49	NA	NA	-
SLN mapping	17,246.01	11.55	−3610.07	0.06	Dominant
C: Concomitant EBRT and chemotherapy as standard treatment for patients with lymph node metastases.					
Lymphadenectomy	22,610.54	11.69	NA	NA	-
SLN mapping	19,011.64	11.76	−3598.89	0.07	Dominant

## Data Availability

Data are contained within the article.

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
