# Peer review of "The Cost-Effectiveness of Sentinel Lymph Node Mapping in High-Risk Endometrial Cancer"

_cancers, 2024, doi:10.3390/cancers16244240_

Round 1

Reviewer 1 Report

Comments and Suggestions for Authors

This is a study in which the authors highlight the cost-effectiveness of sentinel lymph node mapping compared to a routine pelvic lymphadenectomy for the assessment of lymph nodes in patients with high-risk endometrial cancer. Some suggestions are listed below

There is no homogeneity in the typeface.

Figure 1. Include in the figure caption the abbreviation LMN cited in the figure.

Figure 2 I suggest that the box highlighted as adjuvant therapy be inserted with an arrow in the same place where it is mentioned to eliminate the asterisk.

In Table 1. I suggest eliminating “Pooled data” and just leaving the references and representing the text of the columns to the left and not centered.

Eliminate the word “value” and leave only (n) % in separate columns, and eliminate 95%CI and leave only the word “range”.

In Table 2 column corresponding to “ICER (€/QALY)” the sign (-) means that the data or deficit does not exist, I suggest clarifying, and if the data does not exist, I suggest eliminating this column.

It is an important study where it proposes a decision analysis model to compare SLN mapping with pelvic lymphadenectomy to guide adjuvant treatment in patients with high-risk endometrioid and non-endometrioid SC in terms of costs and effects.

Author Response

Reviewer: This is a study in which the authors highlight the cost-effectiveness of sentinel lymph node mapping compared to a routine pelvic lymphadenectomy for the assessment of lymph nodes in patients with high-risk endometrial cancer. Some suggestions are listed below.

Comment 1: There is no homogeneity in the typeface.
Response 1: The lay-out of the current manuscript is in Cancers’ house style. If we need to make some adjustments, we would be happy to hear.

Comment 2: Figure 1. Include in the figure caption the abbreviation LMN cited in the figure.
Response 2: Thank you for pointing this out. We included this, and some other abbreviations, as suggested.

Comment 3: Figure 2 I suggest that the box highlighted as adjuvant therapy be inserted with an arrow in the same place where it is mentioned to eliminate the asterisk.
Response 3: We understand the reviewers comment, however we do believe that this does not make the figure more clear. But if this should be adjusted, we would love to hear it.

Comment 4: In Table 1. I suggest eliminating “Pooled data” and just leaving the references and representing the text of the columns to the left and not centered.
Response 4: We agree with this comment and thank the reviewer for this. We changed this as suggested by the reviewer.

Comment 5: Eliminate the word “value” and leave only (n) % in separate columns, and eliminate 95%CI and leave only the word “range”.
Response 5: We agree that this simplifies the table, therefore we eliminated the words ‘value’ and the ‘range’.

Comment 6: In Table 2 column corresponding to “ICER (€/QALY)” the sign (-) means that the data or deficit does not exist, I suggest clarifying, and if the data does not exist, I suggest eliminating this column.
Response 6: We thank the reviewer for his attentiveness. We eliminated the column, as no ICER was calculated since SLN mapping appeared to be the dominant strategy, with both lower costs and higher QALYs compared to lymphadenectomy.

It is an important study where it proposes a decision analysis model to compare SLN mapping with pelvic lymphadenectomy to guide adjuvant treatment in patients with high-risk endometrioid and non-endometrioid SC in terms of costs and effects.
Comment: We thank the reviewer for his kind words.

Reviewer 2 Report

Comments and Suggestions for Authors

I particularly appreciated the study, which is very timely in this era of resource scarcity. The study is conducted correctly and takes into account the costs of each procedure. If I have to make a comment it concerns the execution system of the SLN mappimg. In this study, only the indiocyanine method is taken into account. It would have been useful to preliminarily take into consideration also the lymphoscintigraphic method in relation to costs, but also the comparison of the sensitivity and specificity of the two methods.

Author Response

Reviewer: I particularly appreciated the study, which is very timely in this era of resource scarcity. The study is conducted correctly and takes into account the costs of each procedure.

Comment 1: If I have to make a comment it concerns the execution system of the SLN mappimg. In this study, only the indiocyanine method is taken into account. It would have been useful to preliminarily take into consideration also the lymphoscintigraphic method in relation to costs, but also the comparison of the sensitivity and specificity of the two methods.

Response 1: We thank the reviewer for his compliments. We understand his suggestion on comparing multiple techniques of SLN mapping (ICG versus lymphoscintigraphy with patent blue). However, literature shows SLN mapping with ICG to be the preferred method. We, therefore, only included ICG as tracer for SLN mapping. We highlighted this in out method and discussion section.

Reviewer 3 Report

Comments and Suggestions for Authors

A holistic approach to treatment decision-making, modeling cost-effectiveness in terms of quality-adjusted life years. An example of well-written work

Author Response

Reviewer: A holistic approach to treatment decision-making, modeling cost-effectiveness in terms of quality-adjusted life years. An example of well-written work.

Response: We are pleased to read the words of the reviewer and we would like to thank the reviewer for his time and effort.

Reviewer 4 Report

Comments and Suggestions for Authors

Thank you for the opportunity to review this paper.

The research uses a decision tree model to evaluate the cost utility of SNL versus lymphadenectomy for EC.

A full and clear description of the intervention and comparator would be useful under the methods section. i.e. what does each procedure involve, outpatient/inpatient, equipment, duration?

A one-year cycle has been chosen and this needs to be explained and justified in detail.

The model is most sensitive to the sensitivity of SNL (97%). The paper by Rossi presents the sensitivity and specificity in Table 3. The limitations of this value should be discussed in detail in the discussion (sample size etc).

Some parameters used expert opinion as their source. This needs to be described in detail, (even if not named), for example the expert’s role, what they commented on and when.

The utility of 1 has been used for “no recurrence, no toxicity”. This is not accurate. Utility values from the general (well) population are less than 1. The highest utility should at most equal the general (Dutch) population.

Death is recorded as utility of 1. This is an error. Is it a typo or is this the value in the model?
The cost perspective is the Netherlands and should be clearly stated.

Comments on the Quality of English Language

The manuscript will benefit from some English language editing, checking for typos and checking font.

Author Response

Reviewer: The research uses a decision tree model to evaluate the cost utility of SNL versus lymphadenectomy for EC.

Comment 1: A full and clear description of the intervention and comparator would be useful under the methods section. i.e. what does each procedure involve, outpatient/inpatient, equipment, duration?Response 1: We agree with the reviewer that we should extend the description of both the pelvic lymphadenectomy and SLN mapping. Therefore, we added a part on describing both interventions in the Methods section, sub-section ‘Model structure’.  We hope this is according to the reviewers wishes.

Comment 2: A one-year cycle has been chosen and this needs to be explained and justified in detail.
Response 2: We added a part on this in our method section (‘Model structure’): the cycle duration in the Markov model was one year, since this aligns well with the natural disease progression. 

Comment 3: The model is most sensitive to the sensitivity of SNL (97%). The paper by Rossi presents the sensitivity and specificity in Table 3. The limitations of this value should be discussed in detail in the discussion (sample size etc).
Response 3: We agree with the reviewer that using only one source for the sensitivity of SLN mapping might be a limitation, however, other studies (including systematic reviews and meta-analyses) show comparable sensitivity rates. We added a part on this in our Discussion section.

Comment 4: Some parameters used expert opinion as their source. This needs to be described in detail, (even if not named), for example the expert’s role, what they commented on and when.
Response 5: We thank the reviewer for pointing this out. Accordingly, we added a clarification on this throughout the manuscript.

Comment 5: The utility of 1 has been used for “no recurrence, no toxicity”. This is not accurate. Utility values from the general (well) population are less than 1. The highest utility should at most equal the general (Dutch) population.
Response 6: We added a part (including reference) in the Discussion section: For the health state ‘no recurrence, no toxicity’ we assumed a utility value of 1, whereas actually the mean utility for age matched participants in the general population is lower than 1. The utility value of 1 represents the best possible health state within this studies context, even though it might slightly exceed typical general population health values. Using a utility of 1 for this health state allows the model to clearly capture the relative impact of being disease-free versus experiencing disease progression or treatment side effects.

Comment 7: Death is recorded as utility of 1. This is an error. Is it a typo or is this the value in the model?
Response 7: We thank the reviewer for his bright eye, this should be a 0 indeed and we changed this in the table accordingly.

Comment 8: The cost perspective is the Netherlands and should be clearly stated.
Response 8: This is a valuable remark of the reviewer. In the Method section, we clarified that the study was conducted form a Dutch health care perspective and we added a part on the applicability of our study for other countries in the Discussion section.

Round 2

Reviewer 4 Report

Comments and Suggestions for Authors

Revisions accepted.